# OpenReview forum: "Reasoning Models Sometimes Output Illegible Chains of Thought"
_NeurIPS.cc/2025/Conference — NeurIPS 2025 poster_

### Official Review · Reviewer_sJFd · 2025-06-12

**Clarity:** 3
**Significance:** 2
**Originality:** 3
**Rating:** 3
**Confidence:** 5

**Summary:**

This paper investigates an intriguing issue with large language models trained with outcome-based reinforcement learning (RL). These models often generate illegible CoTs that include nonsensical phrases, language mixing, and fabricated words, yet somehow lead to coherent and correct final answers. Interestingly, the authors find that more illegible CoTs are often associated with better performance, especially on harder questions.
The authors examine this behavior through empirical evaluation on the GPQA-Diamond dataset, use Claude 3.5 Sonnet as an automated legibility grader, and compare different models (including Claude 3.7 Sonnet and DeepSeek V3) across difficulty levels.

**Questions:**

**Questions**

1. The authors mention that Claude 3.7 exhibits more legible CoTs compared to DeepSeek R1, despite also being trained with outcome-based RL. Can you please elaborate on what differentiates Claude’s training or architecture that might explain this behavior?
2. The 1–9 legibility scoring system is central to your evaluation, but its criteria remain somewhat vague. Could you provide a clearer definition or rubric that distinguishes between scores?
3. Including one or two representative prompts in the main paper would help readers better understand your experimental setup.
4. Since temperature affects generation randomness, it would be helpful to see a more systematic analysis of how varying temperature impacts CoT legibility and correctness. This is critical for understanding robustness.

5. Have you considered removing illegible segments from CoTs and observing whether the final answer remains unchanged? This could help determine whether these segments are actually contributing to reasoning or are merely artifacts.

2. The authors mentions some disagreement between Claude and human annotators in legibility grading.  A more detailed analysis of these discrepancies (e.g., inter-rater reliability or specific examples) would strengthen the case for relying on model-based grading.

**Ethical Concerns:**

["NO or VERY MINOR ethics concerns only"]

**Final Justification:**

While the authors addressed several concerns and promised improvements, the fundamental limitation remains unchanged. This is primarily a descriptive study without rigorous hypothesis testing. I maintain my previous score.

**Limitations:**

Yes

**Quality:**

2

**Strengths And Weaknesses:**

**Strengths**
1. The authors identified a very critical concern that illegible reasoning before a correct answer can potentialy harm interpretability and trust. I liked the reasoning around steganography.
2. The authors made GPQA dataset harder by removing answer choices, which is a good design choice.
3. Thoughtful, well-scoped hypotheses are proposed.
4. The limitation section is honest and specific — it does not gloss over flaws or oversell results.
5. There’s a valuable AI safety angle here: if illegibility increases with question difficulty, then CoT monitoring may be least reliable when we need it the most.

**Weakness**
1. The biggest limitation is that the paper is mostly descriptive. It doesn’t test its main hypotheses.
2. The term “illegibility” is used a lot but is never clearly defined. Is it syntactic noise? Semantic incoherence? Non-English tokens? A more intuitive or formal breakdown would help.
3. The meaning of each score on the 1–9 scale is never fully explained. What makes a 3 vs. a 5? Are there scoring rules or rubrics? This makes the interpretability of their figures difficult.

3. While prompts are included in the supplementary code, it would help to show one or two in the paper for clarity and reproducibility.
5.  The model comparison would be more robust if an open model (e.g., LLaMA or Mistral) were included.
6.  The correlation between illegibility and correctness is interesting but not necessarily meaningful—is illegibility actually helping reasoning? Or is it just correlated because of some third factor like model uncertainty or longer outputs?

---

> ### Author Rebuttal · Authors · 2025-07-31
>
> We are thankful for your time and helpful feedback. We were glad to hear that you liked the hypotheses proposed!
>
> > The biggest limitation is that the paper is mostly descriptive. It doesn’t test its main hypotheses.
>
> We agree this would be beneficial. We did not explore more hypotheses in detail earlier because we thought the primary impact of this paper to be its relevance to chain-of-thought monitoring, for which a demonstration of an increasingly salient problem seemed useful even if we couldn’t explain its mechanism with sufficient rigor. Chain of thought monitoring is increasingly prominent in frontier AI safety work (e.g. DeepMind and OpenAI both consider it to be a major part of their safety agenda), so we consider it especially salient right now.  Given more time since the submission, we have been working on more experiments now for understanding the mechanism here.
>
> To specify our current understanding, we currently think that this is a consequence of token importance in chains-of-thought being very heavy-tailed. There is some recent work supporting this: for example, a paper showing that the tokens with highest entropy in a CoT are responsible for most of the gains from RL such that masking gradient updates to all but those tokens often outperforms models trained in the standard way, and mechanistic interpretability work showing a small pool of sentences receiving disproportionate attention from future sentences. This is related to the vestigial reasoning hypothesis we mention in the paper, and we are working on analyzing entropy distributions in legible and illegible CoTs.
>
> > The term “illegibility” is used a lot but is never clearly defined. Is it syntactic noise? Semantic incoherence? Non-English tokens? A more intuitive or formal breakdown would help.
>
> We intend the term to be used pragmatically, in reference to how easy it is for a monitor to understand the contents of the text. This could take the form of syntactic noise, semantic incoherence, language mixing, and more. We will add a clearer definition in the final version of the paper.
>
> > The meaning of each score on the 1–9 scale is never fully explained. What makes a 3 vs. a 5? Are there scoring rules or rubrics? This makes the interpretability of their figures difficult.
>
> We few-shot prompt the autograder with examples of human-graded text, along with a description of how to judge the text. The full prompt is in the code in the supplementary material, but we will add it to the appendix of the paper. There are examples of text scoring 3 and 5 in Table 1.
>
> > While prompts are included in the supplementary code, it would help to show one or two in the paper for clarity and reproducibility.
>
> We fully agree; as mentioned we will add this as an appendix to the final version of the paper.
>
> > The model comparison would be more robust if an open model (e.g., LLaMA or Mistral) were included.
>
> We will add results from more models to the paper, as detailed in response to reviewer Y9nr above.
>
> > The correlation between illegibility and correctness is interesting but not necessarily meaningful—is illegibility actually helping reasoning? Or is it just correlated because of some third factor like model uncertainty or longer outputs?
>
> We agree this is hard to rigorously establish. We plan to add some new analysis based on what the upstream factor could be (token importance being heavy-tailed), as mentioned above. That said, the main claim is primarily that outcome-based RL can lead to more illegibility, not necessarily that it’s a straightforward causal relationship. The implications for AI safety and monitoring are similar in both cases, though we agree plans for mitigation would look quite different.
>
> > The authors mention that Claude 3.7 exhibits more legible CoTs compared to DeepSeek R1, despite also being trained with outcome-based RL. Can you please elaborate on what differentiates Claude’s training or architecture that might explain this behavior?
>
> We’re unsure. Claude’s model card doesn’t reveal training details like that. The paper has some speculation on potential interventions that could have caused this, based on other work Anthropic has done on preserving reasoning oversight (e.g. paraphrasing and re-training).
>
> > The 1–9 legibility scoring system is central to your evaluation, but its criteria remain somewhat vague. Could you provide a clearer definition or rubric that distinguishes between scores?
>
> Our current strategy is primarily few-shot prompting with human-graded text, along with a general guide for rating text based on how legible it appears to the model. We considered a more precise rubric, but didn’t want to bias the models too much toward one kind of illegibility—for example, at the start the focus was primarily on language switching since that was mentioned prominently in the R1 paper, but there are many instances of illegible text without language switching. We agree that this should be clearer, and will update the paper with more details about the prompting setup (including an appendix with the prompts themselves).
>
> > Since temperature affects generation randomness, it would be helpful to see a more systematic analysis of how varying temperature impacts CoT legibility and correctness. This is critical for understanding robustness.
>
> We agree that this would be useful. Our testing shows that most temperatures below 0.9 result in behavior similar to temperature 0 for R1, but not for R1-Zero. This could suggest that the primary effect from the added legibility training for R1 was on lower temperatures. We will update the paper with these results.
>
> > Have you considered removing illegible segments from CoTs and observing whether the final answer remains unchanged? This could help determine whether these segments are actually contributing to reasoning or are merely artifacts.
>
> This was an experiment we were very excited to test, but no API provider seems to support this kind of pre-filling with a forced answer. In our tests, the model would ignore such a pre-fill and generate a new one from scratch. We couldn’t test a locally hosted version of the model because of compute constraints.
>
> > The authors mentions some disagreement between Claude and human annotators in legibility grading. A more detailed analysis of these discrepancies (e.g., inter-rater reliability or specific examples) would strengthen the case for relying on model-based grading.
>
> Late into writing this paper, we noticed that our autograder wasn’t well-designed to pick up on whether models were deciding on their final answer within the legible portion of the CoT. Owing to time constraints in setting up and testing a better autograder, we instead had human contractors judge a random 5% of samples, and added it into the paper at the last minute.
>
> We agree that this is poorly explained. We also think that the number of samples tested were too low, so we will improve the autograder and update the paper with revised results.
>
> Thank you again for taking the time to review the paper and providing helpful feedback! Do the above actions address your concerns with the paper? If not, what further clarification or modifications could we make to improve it?

---

> > ### Comment · Reviewer_sJFd · 2025-08-05
> >
> > Thanks for the detailed and thoughtful rebuttal. I appreciate the clarifications. That said, I still feel the paper is largely descriptive. The central idea is intriguing, but without directly testing that hypothesis, it’s hard to evaluate how much to take away from the results. I will maintain my score as before.

---

### Official Review · Reviewer_Cvn4 · 2025-06-30

**Clarity:** 3
**Significance:** 2
**Originality:** 1
**Rating:** 3
**Confidence:** 5

**Summary:**

This paper investigates the legibility issue of reasoning traces generated by recent large reasoning models. By prompting Claude-3.5 to evaluate CoT outputs from DeepSeek-R1, R1-Zero, DeepSeek-V3, and Claude 3.7 Sonnet on the GPQA dataset, the authors observe that DeepSeek-R1 and R1-Zero frequently produce illegible reasoning traces. Notably, they find a positive correlation between CoT illegibility and answer correctness, suggesting that illegible reasoning might be beneficial for performance.

**Questions:**

In Figure 3, why was temperature = 0 (greedy decoding, although with potential minor variations in practice due to API implementation and hardware-specific issues) chosen, along with multiple generations, to determine question hardness?

**Ethical Concerns:**

["NO or VERY MINOR ethics concerns only"]

**Final Justification:**

This paper studies an important and emerging topic in recent reasoning models. However, my main concern lies in the overall quality of the work. Specifically, both I and other reviewers have raised issues regarding the size of the test set, the number of evaluated models, and the rigor of the behavioral analyses. Even during the rebuttal phase, the authors did not present any concrete or persuasive additional results, which reinforces my belief that the paper is not suitable for acceptance at NeurIPS and should be rejected. Therefore, I am maintaining my score.

**Limitations:**

See weakness above

**Paper Formatting Concerns:**

No paper formatting issue.

**Quality:**

1

**Strengths And Weaknesses:**

**Strength:**
- This paper studies an emerging and significant issue within large reasoning models.
- The identified positive correlation between the correctness of answers and the illegibility of reasoning traces is intriguing and presents potential implications for monitoring and interpreting model behaviors.

**Weakness:**
- The evaluation is considerably limited. Using only the GPQA diamond dataset, which contains just 198 questions, is insufficient to draw persuasive generalizations about model behavior.
- The paper restricts its analysis to only two families of models (DeepSeek and Claude). A more comprehensive evaluation across a broader set of reasoning models, such as QwQ and Qwen3, and DeepSeek distilled models, would strengthen the study and enhance generalizability.
- While the paper emphasizes evaluating the impact of temperature, it considers only two extreme temperature values (0 and 1), neglecting other practically relevant settings. For example, DeepSeek-R1’s default generation temperature is 0.6 [1], which is notably absent from the study.
- The analysis provided is largely shallow and observational, lacking systematic, hypothesis-driven experimentation. Given the expectations for top venues such as NeurIPS, it would be beneficial to clearly formulate hypotheses and conduct controlled experiments to rigorously test and validate them.
- Experimental details are inadequately documented. Crucial information such as the exact prompts used for the Claude-3.5 judging process is not provided, limiting reproducibility and clarity.

[1] DeepSeek-AI DeepSeek-R1: Incentivizing Reasoning Capability in LLMs via Reinforcement Learning

---

> ### Author Rebuttal · Authors · 2025-07-31
>
> We thank you for your thoughtful feedback.
>
> > The evaluation is considerably limited. Using only the GPQA diamond dataset, which contains just 198 questions, is insufficient to draw persuasive generalizations about model behavior.
>
> We will add results from more datasets to the paper, as detailed in response to reviewer Y9nr above.
>
> > The paper restricts its analysis to only two families of models (DeepSeek and Claude). A more comprehensive evaluation across a broader set of reasoning models, such as QwQ and Qwen3, and DeepSeek distilled models, would strengthen the study and enhance generalizability.
>
> We will add results from more models to the paper, as detailed in response to reviewer Y9nr above.
>
> > While the paper emphasizes evaluating the impact of temperature, it considers only two extreme temperature values (0 and 1), neglecting other practically relevant settings. For example, DeepSeek-R1’s default generation temperature is 0.6 [1], which is notably absent from the study.
>
> We studied various intermediate temperature settings, but found the results to be very similar to the setting with temperature 0. We will include this in the final version of the paper (along with other generation settings such as top_p, which we also found variance on).
>
> We note though that the DeepSeek API docs for R1 recommend using a temperature of 1 by default.
>
> > The analysis provided is largely shallow and observational, lacking systematic, hypothesis-driven experimentation. Given the expectations for top venues such as NeurIPS, it would be beneficial to clearly formulate hypotheses and conduct controlled experiments to rigorously test and validate them.
>
> We agree this would be beneficial. We did not explore more hypotheses in detail earlier because we thought the primary impact of this paper to be its relevance to chain-of-thought monitoring, for which a demonstration of an increasingly salient problem seemed useful even if we couldn’t explain its mechanism with sufficient rigor. Chain of thought monitoring is increasingly prominent in frontier AI safety work (e.g. DeepMind and OpenAI both consider it to be a major part of their safety agenda), so we consider it especially salient right now. Given more time since the submission, we have been working on more experiments now for understanding the mechanism here.
>
> To specify our current understanding, we currently think that this is a consequence of token importance in chains-of-thought being very heavy-tailed. There is some recent work supporting this: for example, a paper showing that the tokens with highest entropy in a CoT are responsible for most of the gains from RL such that masking gradient updates to all but those tokens often outperforms models trained in the standard way, and mechanistic interpretability work showing a small pool of sentences receiving disproportionate attention from future sentences. This is related to the vestigial reasoning hypothesis we mention in the paper, and we are working on analyzing entropy distributions in legible and illegible CoTs.
>
> > Experimental details are inadequately documented. Crucial information such as the exact prompts used for the Claude-3.5 judging process is not provided, limiting reproducibility and clarity.
>
> The prompts and code for replicating the results are provided in the supplementary material. We agree that it would be much better to include the prompts as an appendix in the paper, and will add that to the final version of the paper.
>
> > In Figure 3, why was temperature = 0 (greedy decoding, although with potential minor variations in practice due to API implementation and hardware-specific issues) chosen, along with multiple generations, to determine question hardness?
>
> Setting the temperature to 0 was primarily just to account for variance in model response. “Multiple generations” was poor phrasing—it’s meant to indicate that the question responses were sampled separately from R1 at temperature 0. We will edit that to be clearer.
>
> We thank you again for taking the time to review the paper and providing helpful feedback! Do the above actions address your concerns with the paper? If not, what further clarification or modifications could we make to improve it?

---

> > ### Comment · Reviewer_Cvn4 · 2025-08-01
> >
> > Thank you for your response. However, most of my points are not adequately addressed in the rebuttal, particularly due to the lack of concrete additional results and the limited rigor in the behavior analysis, as also noted by other reviewers. Thus, I will maintain my score. (P.S. The default and fixed temperature setting for DeepSeek-R1 is 0.6, as clearly stated in its Huggingface model card.)

---

> > > ### Author Response · Authors · 2025-08-03
> > >
> > > Thank you for your response. We would be happy to provide quantitative results; because of the lack of PDF support, we are somewhat limited in how we can present them in the replies. Here are some of the new results as a table (this is not exhaustive):
> > >
> > > | Model, Dataset | Mean | Standard Deviation | % scoring above 3 | % scoring above 5 | % scoring above 7 |
> > > |-------|------|-------------------|-----------|-----------|-----------|
> > > | DeepSeek R1, GPQA-Diamond | 4.5 | 2.5 | 70% | 45% | 29% |
> > > | R1-Distill Llama 70B, GPQA-Diamond | 2.2 | 1.0 | 38% | 3% | 0% |
> > > | R1-Distill Qwen 32B, GPQA-Diamond | 2.3 | 1.1 | 40% | 4% | 0% |
> > > | QwQ, GPQA-Diamond | 2.9 | 1.7 | 43% | 16% | 6% |
> > > | DeepSeek R1, MMLU-Pro | 2.1 | 1.6 | 24% | 7% | 4.5% |
> > > | DeepSeek R1, MMLU-Pro (only traces > 10000 chars) | 3.8 | 2.1 | 73% | 25% | 17% |
> > > | DeepSeek R1, ChemBench | 5.9 | 2.5 | 87% | 68% | 55% |
> > >
> > > As mentioned above, R1-distill Llama 70B and R1-distill Qwen 32B have notably legible traces. Since they were not trained with RL, this could be seen as evidence toward illegibility being a cause of RL in particular. QwQ's scores are higher than any non-RL model, though slightly lower than R1. We included the MMLU-Pro subset with reasoning traces longer than 10000 characters here because many of the questions are too easy and have much shorter answers than the other datasets.
> > >
> > > We would be happy to provide more rigorous results from hypothesis testing, if that is what you're referring to with behavior analysis. As mentioned earlier however, we think that empirical documentation of these patterns is what is primarily of direct relevance to chain of thought monitoring, which motivated rapid dissemination given the pace of deployment.
> > >
> > > Thank you for pointing out the temperature recommendation on the Huggingface model card. This is interesting, since the DeepSeek API docs still recommend a temperature of 1, which is what we were referring to—though it's possible that this is simply a bug.

---

> > > > ### Comment · Reviewer_Cvn4 · 2025-08-05
> > > >
> > > > Thank you for your additional response with detailed performance results. It is interesting to see that QwQ, despite its much smaller size, produces significantly fewer illegible outputs than the much larger DeepSeek R1. I will raise my score to 3, given these additional evaluation results.

---

### Official Review · Reviewer_kSGx · 2025-06-30

**Clarity:** 4
**Significance:** 3
**Originality:** 3
**Rating:** 5
**Confidence:** 4

**Summary:**

This paper investigates CoT tokens in reasoning models (specifically R1, R1-Zero, V3, and Claude 3.7 Sonnet).

The authors used a particular data-set of scientific questions (GPQA-Diamond). They examined: 1) whether the models got the answers correct or incorrect, 2) the degree of 'illegibility' in model answers, whether in the 'Response' or 'Reasoning' sections.

The authors measured illegibility by prompting another model (Claude 3.5) to use a 1-9 illegibility scale.

The authors found a high-degree of illegibility in the 'reasoning' sections of R1 and R1-Zero. The response sections, and V3 and Claude 3.7 Sonnet have low illegibility. The authors conclude that outcome-based RL leads to illegible CoTs, and propose several possible hypotheses for why this might be the case, working through how likely they are.

The authors conclude by mentioning several limitations of their work, including the testing of a limited dataset, limited models, and the need for controlled experiments to determine causal mechanisms.

**Questions:**

Questions and suggestions are spelled out above in the 'weaknesses' section. While I strongly suggest the authors address the issues under "ALSO", the most important thing that would make this a clearly significant paper with solid impact are the three things the authors themselves list in limitations:

1. Moving beyond the specific data-set, to help generalize the conclusions
2. Increasing the number of models, to help generalize the conclusions
3. Testing *some* of their hypothesis mentioned in the discussion in a clear way that would allow to meaningfully conclude something about why this phenomenon is happening and when. I'm not asking for a full-blown worked-out theory.

**Ethical Concerns:**

["NO or VERY MINOR ethics concerns only"]

**Final Justification:**

Several of my previous issues have been resolved, and so I am updating the score.

**Limitations:**

I'm not sure what is meant by 'addressed'. The authors have adequately _brought up_ the limitations of their work, and they are absolutely commended for it. The strongest weaknesses of the paper are themselves mentioned by the authors. They do not 'address' them in the sense that they don't handle these weaknesses yet.

**Quality:**

3

**Strengths And Weaknesses:**

Strengths
=======

This paper is well-written, and timely. It addresses an important issue that many people in the field would find of interest, under the general umbrella of "what exactly is the relationship between the overt tokens in CoTs of reasoning models, and the actual reasoning?". The authors also reasonably identify limitations with their own work.

As far as originality goes, I make no strong claims on saying this is highly original given that there's just so much work out there, but at least for my part I'm not familiar with something quite like it.

Weaknesses
==========

MAIN:
-------
The authors do a good job of identifying their own weaknesses in the 'limitations' section, and I agree with them there, namely:

* The use of a limited data-set, one that is also particularly difficult in its use of scientific jargon, makes it hard to generalize the conclusions
* The authors used a limited set of reasoning models, and even for those, 2 of the 4 didn't follow the conclusions they draw
* There is a lack of causal mechanism or clear explanation for why and when we observe this illegibility. The authors put forth several reasonable hypotheses in the discussion, but again they themselves point out the need for further testing, ablation studies, and so on to determine what is actually going on. This makes the paper of limited impact, and I mostly just agree with the authors' own assessment.

ALSO
-------

The following is a list of issues that I had as I read the paper, they are in order of reading rather than importance, so some of these are quite minor. All of them are given with the aim of helping to improve the paper:

1. "we study whether outcome-based RL causes reasoning models to do meaningful reasoning in illegible CoT" -- this is a weird sentence, and it is unclear what you mean. Especially as this is THE most important sentence in the opening ("we study X"). What is 'meaningful' reasoning? How do you do reasoning "IN" CoT? Based on your conclusions, you also don't study that.

2. Claude 3.5 as the auto-grader: I think the authors have a reasonable justification for this, but it would still be helpful to see what the actual prompting for Claude here was. In comparing to o1 as the autogravder, the authors mention "an average difference in score of 0.44±0.78" -- it wasn't clear to me how this comparison was done. Did you simply take Mean(sum(score(Claude) - score(o1)))? If so, you could be missing huge discrepancies. Or did you take the absolute value? In general it seems better to use techniques developed for inter-rater agreement here like Cohen's kappa. I don't think any of this will change the conclusions, but still.

3. "we find a positive correlation between CoT illegibility and accuracy of model answers." -- that's not technically true. You have a binary split between correct and incorrect answers, and your statistical test (not detailed, see below) finds that the legibility for these two partitions is statistically different. That is not the same as a correlation of illegibility and accuracy.

4. "These results suggest that illegible reasoning is likely a consequence of outcome-based RL" -- why does it suggest that, just based on this graph? And what is each dot representing here?

5. Figure 6: ""We then plot the distribution of these scores across all the CoTs. The plot shows a clear trend of CoTs becoming more illegible as they progress." -- by what statistical test? I don't doubt you're right, but a specific claim like that requires an actual test to back it up.

6. "This suggests that lengthy CoTs may be detrimental to CoT monitoring, especially as medium legibility scores are more clear
condensed reasoning and replicate across models, before progressing to stronger illegibility." -- minor issue, but the middle part of that sentence makes no sense.

7. "This stark contrast with DeepSeek and OpenAI models suggests that the illegibility we observe is not an inevitable consequence of outcome-based RL, but at most a likely consequence." -- Do you mean Anthropic models? I thought you tested Claude, not o1/3. Also, it is possible they DID encounter this (that is, that it *is* an inevitable consequence) but found additional ways around it once it cropped up, no?

8. "For CoT legibility evaluation, we think it’s useful to be pretty lenient — in realistic monitoring setups, we will likely assist the monitor" -- This whole sentence is odd. Lenient in what sense? Who is the 'we' that will be assisting the monitor? Assisting them how?

9.  "In our tests, Claude’s responses often differed from human evaluators on this specific question."  -- what human evaluators? what tests? The paper is written as though there are no human evaluators, and no further information is provided about this.

10. " For these cases, we stick with human evaluation on a smaller set of samples (5%) instead of the automated scoring. - which cases? how do you know which cases these are? Who are the humans doing this evaluation, and what is their metric? What is their agreement?

11. "Text scoring 3 or higher often exhibits condensed reasoning, with articles and common words being dropped and
sentences being cut short." -- is there an actual analysis to go with this statement? Right now it is extremely subjective and qualitative.

12. " the Claude autograder sometimes becomes confused by the content of such CoTs.  -- very inexact, what does it mean 'become confused' and 'sometimes'?

13. "We find that R1 and R1-Zero both have significantly more illegible CoTs when answering harder questions (Figure 3), with an average relative increase of 23% for R1 and 29% for R1-Zero." -- what is the statistical basis for this statement? increase in 23% from what to what? From the easiest to the hardest? Eye-balling Figure 3 I don't think there's a statistical difference between the easy and hard cases. I'm willing to be convincing otherwise but in the absence of an actual test I can't accept the statement or the conclusions that follow from it.

14. Figure 4: What is 'point density' meant to be?

---

> ### Author Rebuttal · Authors · 2025-07-31
>
> Thank you for the thoughtful feedback, especially the detailed comments! We’d like to address the weaknesses and questions you brought up.
>
> > The use of a limited data-set, one that is also particularly difficult in its use of scientific jargon, makes it hard to generalize the conclusions
>
> We plan to add results from more datasets to the paper, as detailed in response to reviewer Y9nr above.
>
> > The authors used a limited set of reasoning models, and even for those, 2 of the 4 didn't follow the conclusions they draw
>
> We are testing more reasoning models, such as the Qwen and R1-distill models, and will add those results to the paper.
>
> However, we note that DeepSeek-V3, one of the models tested in the paper, is not a reasoning model, and was included to compare against R1 and R1-Zero, models trained with RL from V3. Claude 3.7 Sonnet does not have illegible CoTs, but we don’t believe this contradicts our conclusions—we mention that it suggests that what we observe in R1 and R1-Zero is not a necessary consequence of RL, but at most a likely consequence. New results from other reasoning models (detailed in our response to reviewer Y9nr above) suggest this to be the case as well. Our conclusions are that RL poses risk to monitoring ability in some cases, not that it always does so.
>
> > There is a lack of causal mechanism or clear explanation for why and when we observe this illegibility. [...] I mostly just agree with the authors' own assessment.
>
> We agree that this is a very valuable addition; for more information, refer to our response to reviewer Cvn4 below.
>
> > "we study whether outcome-based RL causes reasoning models to do meaningful reasoning in illegible CoT" -- this is a weird sentence, and it is unclear what you mean. Especially as this is THE most important sentence in the opening ("we study X"). What is 'meaningful' reasoning? How do you do reasoning "IN" CoT? Based on your conclusions, you also don't study that.
>
> By ‘meaningful’ reasoning, we meant whether a model is using an illegible CoT in forming its answer. For example, if the model were to output a series of random words after deciding on its final answer verbally, we would consider those words to not have been used for deciding the answer. We tried to study this by looking at whether models decide on their answer before or after illegible tokens arise, and whether they’re more likely to get correct answers with illegible CoTs, but we agree this is correlational and not causal evidence. We will edit this sentence (and other mentions of this clause) to be clearer.
>
> > Claude 3.5 as the auto-grader: I think the authors have a reasonable justification for this, but it would still be helpful to see what the actual prompting for Claude here was. [...] I don't think any of this will change the conclusions, but still.
>
> Prompts were included in the supplementary code, but we agree it would have been a better choice to include examples in an appendix to the paper itself.
>
> We did take mean(sum(score(Claude) - score(GPT-4o))) (our comparison model was GPT-4o and not o1), which we agree is limiting. We get a Cohen’s weighted kappa of 0.782, which we will edit the paper to mention instead.
>
> > "we find a positive correlation between CoT illegibility and accuracy of model answers." -- that's not technically true. You have a binary split between correct and incorrect answers, and your statistical test (not detailed, see below) finds that the legibility for these two partitions is statistically different. That is not the same as a correlation of illegibility and accuracy.
>
> We agree that this was imprecise phrasing. We will edit it to say that correct answers are associated with more illegible CoTs.
>
> > "These results suggest that illegible reasoning is likely a consequence of outcome-based RL" -- why does it suggest that, just based on this graph? And what is each dot representing here?
>
> This is suggested because of the difference between the scores of V3 and R1 / R1-Zero. V3 was the base model that the latter two models were trained from using outcome-based RL (and in R1-Zero’s case, only with outcome-based RL and no supervised fine-tuning). Since V3 has functionally no illegible outputs, this suggests that the illegible reasoning in R1 and R1-Zero arose as a consequence of outcome-based RL. We will update the description to be clearer about this.
>
> The dots represent outlier scores in the distribution. We will update the description to mention this as well.
>
> > Figure 6: ""We then plot the distribution of these scores [...] a specific claim like that requires an actual test to back it up.
>
> We agree that a quantitative metric here would be useful. When computing Spearman’s rank correlation, we get a mean correlation of 0.54 (p <0.0001), and a median correlation of 0.74. We will update the paper to mention this instead.
>
> > "This suggests that lengthy CoTs may be detrimental to CoT monitoring, especially as medium legibility scores are more clear condensed reasoning and replicate across models, before progressing to stronger illegibility." -- minor issue, but the middle part of that sentence makes no sense.
>
> This was a typo, and will be edited. The intended point was that CoTs with medium illegibility scores have stylistic patterns that occur in many other reasoning models (such as o1, or more recently Grok 4), and the fact that we see a continuous shift from fully legible text to this kind of moderately compressed text to fully illegible text might imply that other frontier models may end up with fully illegible text.
>
> > "This stark contrast with DeepSeek and OpenAI models [...] possible they DID encounter this (that is, that it is an inevitable consequence) but found additional ways around it once it cropped up, no?
>
> This was also a typo and will be edited, we did mean Anthropic models.
>
> We agree it’s possible they did encounter it and managed to patch it. We mention some prior work by Anthropic (such as paraphrasing of CoTs) that could explain this behavior in another section of the paper.
>
> > "For CoT legibility evaluation, we think it’s useful to be pretty lenient — in realistic monitoring setups, we will likely assist the monitor" -- This whole sentence is odd. Lenient in what sense? Who is the 'we' that will be assisting the monitor? Assisting them how?
>
> That sentence was intended to explain why we didn’t follow the method in Kirchner et al which uses a time-constrained human contractor for measuring legibility, to simulate a realistic situation with a limited human auditing budget. In practice for CoT monitoring, we think it’s plausible that having some monitoring overhead (in terms of letting the monitor reason at length) and a lack of general knowledge gaps (since model graders have more broad science knowledge than a human contractor and can better evaluate model reasoning) are reasonable affordances AI labs or external auditors will give monitors.
>
> We will update this sentence to be clearer.
>
> > "In our tests, Claude’s responses often differed from human evaluators on this specific question." -- what human evaluators? what tests? The paper is written as though there are no human evaluators, and no further information is provided about this.
>
> > " For these cases, we stick with human evaluation on a smaller set of samples (5%) instead of the automated scoring. - which cases? how do you know which cases these are? Who are the humans doing this evaluation, and what is their metric? What is their agreement?
>
> Since submitting the paper, we’ve decided that relying on human evaluation was a poor choice and will be using an upgraded autograder instead. Refer to the response to reviewer sJFd for more details.
>
> > "Text scoring 3 or higher often exhibits condensed reasoning, with articles and common words being dropped and sentences being cut short." -- is there an actual analysis to go with this statement? Right now it is extremely subjective and qualitative.
>
> We found that samples in this category often have different ways of exhibiting properties of condensed reasoning, such that more specific metrics than the autograder, like words dropped, often doesn’t meaningfully capture what we want. Table 1 shows one excerpt of text with this score; we are planning on adding an appendix with the autograder’s comments on each chunk from this transcript, which have similar analysis.
>
> > " the Claude autograder sometimes becomes confused by the content of such CoTs. -- very inexact, what does it mean 'become confused' and 'sometimes'?
>
> We meant that Claude often describes the text in such CoTs as confusing for it to understand. We will update the description to be clearer about this. Precisely judging the frequency of this is difficult as when asked to explicitly quantify its level of understanding of the text, it often over-claims—e.g. giving high scores even for nonsensical text because the presence of a particular phrase makes it believe it understands the general content.
>
> > "We find that R1 and R1-Zero both have significantly more [...] I can't accept the statement or the conclusions that follow from it.
>
> 23% and 29% represents the increase in the average score on easy questions to hard questions for R1 and R1-Zero. The p-value for the difference between scores for easy and hard questions is 0.0005 for R1’s reasoning, and <0.0001 for R1-Zero’s reasoning.
>
> We will update the paper to include these numbers to justify the claim.
>
> > Figure 4: What is 'point density' meant to be?
>
> Point density is how many samples are close to each other. This was purely for visual clarity since it looked difficult to decipher before. We’ll update the description to better explain this.
>
> Thank you again for the very detailed feedback! It was very useful for improving our paper. Do the above actions address your concerns with the paper? If not, what further clarification or modifications would help to improve it?

---

> > ### Comment · Reviewer_kSGx · 2025-08-04
> >
> > I appreciate the authors' thorough feedback, and am updating from score from a 4 to 5 based on it.
> >
> > I would note that there are several lingering issues here that have not been fully resolved, and that I would advise the authors to address at some point to make their work stronger.
> >
> > Specifically, there is still the use of subjective terms to make technical claims but without backing those up with clear definitions. When asking what the authors mean by 'Claude autograder sometimes becomes confused" (as in, what is 'confused' and what is 'sometimes') the response was "We meant that Claude often describes the text in such CoTs as confusing for it to understand.", a circular and confusing statement that doesn't explain things (ironically), and also "precisely judging the frequency of this is difficult as when asked to explicitly quantify its level of understanding of the text..." means that the authors do not have a way of quantifying 'sometimes'. Such analyses are best either explained or left out.
> >
> > Similar issues apply to the comment right before that about dropped words and such, the authors' response does not clarify the analysis.
> >
> > As something of an aside, I would note that while I do overall appreciate the original work and the proposed revision, I am now a bit concerned about the proposed move to an entirely automated auto-grader rather than dealing with the problems of using human monitors. Consider for example the authors' new statement that they had a problem doing a suggested analysis because the model was "...giving high scores even for nonsensical text because the presence of a particular phrase makes it believe it understands the general content."
> >
> > So it seems that we, as the humans, recognize that certain high scores are unjustified (based on *something* that isn't quantified here). Rather than dig down into what humans are doing here, the suggestion is to replace humans entirely with an auto-grader? By what measure are we measuring 'nonsense', at this point?

---

### Official Review · Reviewer_Y9nr · 2025-07-03

**Clarity:** 3
**Significance:** 3
**Originality:** 3
**Rating:** 4
**Confidence:** 2

**Summary:**

This paper investigates a counterintuitive and underexplored phenomenon: state-of-the-art reasoning language models trained with outcome-based reinforcement learning often generate illegible chain-of-thought reasoning — text that is incoherent, nonsensical, or language-mixed — even when final answers remain accurate. Through evaluation on the GPQA-Diamond dataset, the authors show that illegibility increases with question difficulty and positively correlates with answer correctness. The paper presents hypotheses (e.g., vestigial reasoning, training artifacts) and discusses implications for AI monitoring and safety.

**Questions:**

n/a

**Ethical Concerns:**

["NO or VERY MINOR ethics concerns only"]

**Quality:**

2

**Strengths And Weaknesses:**

S:
1. The paper addresses a highly relevant and novel issue at the intersection of AI safety and interpretability — the emergence of illegible reasoning in high-performing models.

2. The use of clear examples (e.g., Table 1, Figure 1) and progressive CoT breakdown over time makes the findings accessible and convincing.

W:
1. The tested models are limited to Deepseek and Claude, which do not represent the state-of-the-art. How about GPT-o1 and GPT-o3?

2. The focus on a single dataset (GPQA-Diamond) and specific models limits generalizability. It is unclear whether similar results would hold in more naturalistic multi-turn or open-ended reasoning tasks.

---

> ### Author Rebuttal · Authors · 2025-07-31
>
> We are thankful for your time and help. We were glad to hear that you found the paper relevant and convincing!
>
> > The tested models are limited to Deepseek and Claude, which do not represent the state-of-the-art. How about GPT-o1 and GPT-o3?
>
> We agree that it’d be useful to see results from more reasoning models, especially state-of-the-art models such as o1, or o3. Unfortunately, none of the o-series of models provide access to the raw chains of thought; instead, they provide summaries of model reasoning, which likely won’t reveal illegibility and isn’t relevant to our focus. The same applies to the frontier Gemini models.
>
> That said, we acknowledge that testing more models is necessary for understanding how well our results generalize. Since the paper’s submission, we have tested other reasoning models such as the Qwen and R1-distilled models. Preliminary results suggest that QwQ has similar behavior, while the distilled models are noticeably more legible (which would make sense if this is caused by reinforcement learning in particular, since they are solely trained with supervised fine-tuning). We plan to test the new Mistral and Llama-Nemotron reasoning models as well. We will add these results to the final version of the paper.
>
> > The focus on a single dataset (GPQA-Diamond) and specific models limits generalizability. It is unclear whether similar results would hold in more naturalistic multi-turn or open-ended reasoning tasks.
>
> We agree with the importance of testing more settings. We have now tested R1 on more datasets, specifically MMLU-Pro and ChemBench. We find a high variance of scores on MMLU-Pro; on further investigation we find that this is due to many questions being easy and requiring very short reasoning traces. For reasoning traces of length comparable to those from GPQA, we see high illegibility scores. On ChemBench, we find higher illegibility scores than GPQA, partly owing to the questions inducing long reasoning.
>
> We will add these results to the paper, as well as results from testing other models on these new datasets. Are there any datasets you would recommend we consider as well? For example, to better test settings such as multi-turn conversations.
>
> Thank you again for taking the time to review the paper and providing helpful feedback! Do the above actions address your concerns with the paper? If not, what further clarification or modifications could we make to improve it?

---

### Decision · Program_Chairs · 2025-09-17

**Decision:**

Accept (poster)

**Comment:**

This paper presents an intriguing finding with large reasoning models trained with outcome-based reinforcement learning. These models often generate illegible CoTs that include nonsensical phrases, language mixing. Counterintuitively,  illegible CoTs often lead to better accuracy, especially for harder questions. The authors examine this behavior through empirical evaluation on the GPQA-Diamond dataset, use Claude 3.5 Sonnet as an automated legibility grader.

Strengths:
The studied problem is very interesting and timely.

Weaknesses:
1. Limited experiments. Only one small dataset (GPQA-Diamond) is used.
2. Lack of in-depth analysis. this paper is mostly presenting the phenomenon without in-depth analysis of the cause and potential applications.